# Regions within a single epidermal cell of *Drosophila* can be planar polarised independently

**Miguel Rovira†, Pedro Saavedra†‡, José Casal\*, Peter A Lawrence\***

Department of Zoology, University of Cambridge, Cambridge, United Kingdom

**Abstract** Planar cell polarity (PCP), the coordinated and consistent orientation of cells in the plane of epithelial sheets, is a fundamental and conserved property of animals and plants. Up to now, the smallest unit expressing PCP has been considered to be an entire single cell. We report that, in the larval epidermis of *Drosophila*, different subdomains of one cell can have opposite polarities. In larvae, PCP is driven by the Dachsous/Fat system; we show that the polarity of a subdomain within one cell is its response to levels of Dachsous/Fat in the membranes of contacting cells. During larval development, cells rearrange (*Saavedra et al., 2014*) and when two subdomains of a single cell have different types of neighbouring cells, then these subdomains can become polarised in opposite directions. We conclude that polarisation depends on a local comparison of the amounts of Dachsous and Fat within opposing regions of a cell's membrane.

**\*For correspondence:** jec85@
cam.ac.uk (JC); pal38@cam.ac.uk
(PAL)

†These authors contributed
equally to this work

**Present address:** ‡Institute of
Molecular and Cell Biology,
Proteos, Singapore

**Competing interests:** The
authors declare that no
competing interests exist.

**Reviewing editor**:
K VijayRaghavan, National Centre
for Biological Sciences, Tata
Institute for Fundamental
Research, India

## Introduction

Epithelial cells are polarised within the plane of the epithelium and can display consistent orientation across extensive tracts of cells. This property, known as planar cell polarity (PCP) is revealed by polarised structures such as hairs on insect wings or in the skin of mammals. At least two genetic systems are involved in PCP and these are conserved between *Drosophila* and the mouse (*Casal et al., 2006*; *Lawrence et al., 2007*; *Goodrich and Strutt, 2011*). PCP is envisaged as a cellular property, the smallest unit of manifest polarity being an entire single cell. However, we now show in *Drosophila* that different domains within a single cell can have mutually opposing polarities. These multipolar cells occur in the normal larval epidermis and are detectable because the cells are large and some are decorated with several pointed denticles. The Dachsous/Fat (Ds/Ft) system acts at intercellular contacts (*Strutt and Strutt, 2002*; *Ma et al., 2003*; *Casal et al., 2006*); we provide evidence that the polarity of a domain within one cell is its response to the levels of Ds/Ft in neighbouring cells. When another domain of that same responding cell has different neighbours, it can acquire the opposite polarity. We conclude that polarisation of a domain results from a *comparison* of the amounts of Ds and Ft in different regions of the cell membrane. This comparison is made between limited regions of membranes on opposite sides of the same cell that face each other along the anterior to posterior axis. We conjecture that 'conduits' span across the cell and mediate this comparison. In each region of the cell, the orientation of the conduits, a consequence of the comparison, cues the polarity of denticles.

### The later larval stages of *Drosophila*

As we have shown recently (*Saavedra et al., 2014*), the epidermis of the later larval stages of *Drosophila*, the second and third stages, offers considerable advantages for the analysis of PCP. The development of single individuals can be followed and genetic mosaics can be made and studied in vivo (*Saavedra et al., 2014*). The epidermal cells are large (ca 30 µm across in the third stage) and

show their polarity by forming, first, actin-rich predenticles near the cell membrane and then, oriented denticles in the cuticle that they secrete. There are about seven rows of denticulate cells in each segment; denticle rows 0, 1, and 4 point forwards and rows 2, 3, 5, and 6 point backwards (*Figure 1A,B*). There are two rows of muscle attachments called tendon cells, T1 and T2, that lie between rows 1 and 2, and 4 and 5, respectively (*Saavedra et al., 2014*) (*Figure 1*). As the first stage develops into the second, there are neither cell divisions nor cell deaths; nevertheless, the epidermal cells rearrange and some change their identities (*Saavedra et al., 2014*)—for example, the tendon cells form denticles in the embryo and early larva (*Dilks and DiNardo, 2010*) but not in the second and third stages; some cells even change their polarities (*Saavedra et al., 2014*).

## Results and discussion

### Distribution of Ds activity in the segment

Protein interactions between neighbouring cells are at the core of known PCP systems (*Goodrich and Strutt, 2011*). In one of these systems, the protocadherins Ds and Ft form heterodimeric bridges between cells (reviewed in *Lawrence et al., 2007*; *Thomas and Strutt, 2012*). The deployment and orientation of Ds-Ft bridges within different parts of a cell membrane depend on the amounts of Ds and Ft activity in neighbouring cells. For example, a cell with a low level of Ds presents more Ft (than Ds) on its membrane and this draws more Ds (than Ft) to the abutting membrane of the neighbour; thereby affecting the distribution of dimers within the next cell. In this way, the relative numbers and orientations of heterodimers allow a comparison between a cell's anterior and posterior neighbours so that it orients its denticles towards the neighbour with the higher Ds, and/or lower Ft, activity (*Ma et al., 2003*; *Casal et al., 2006*; *Matakatsu and Blair, 2006*).

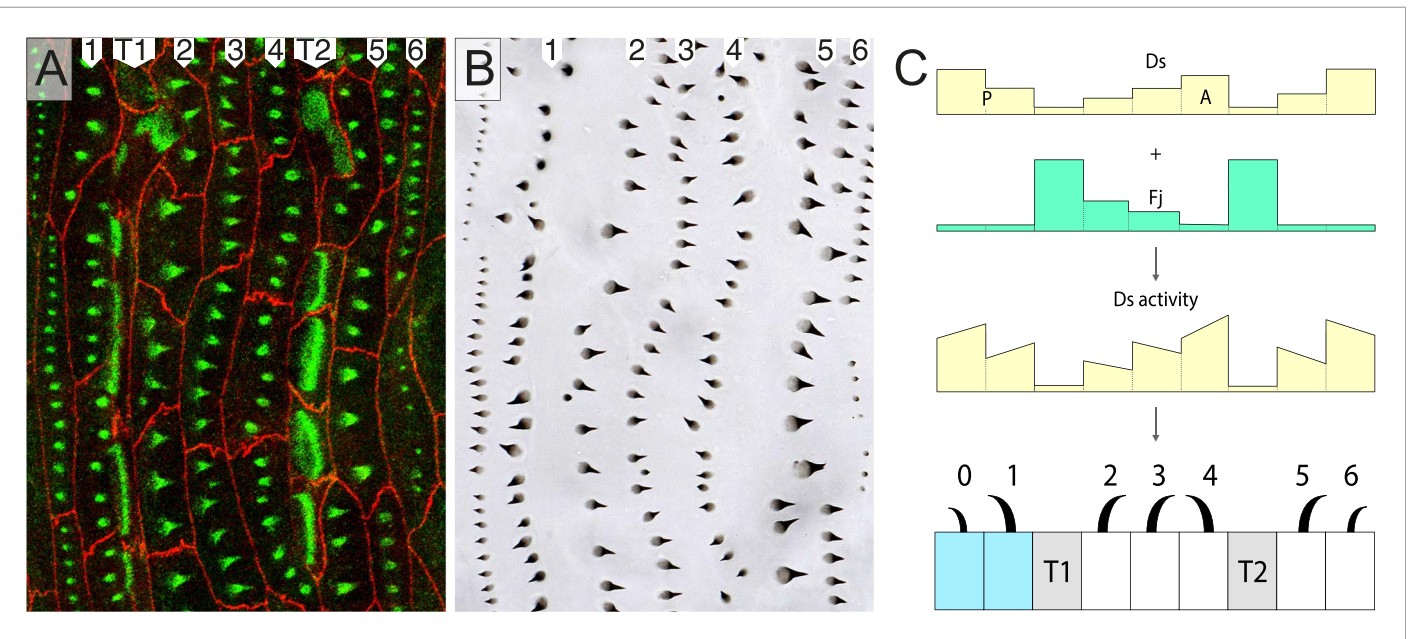

**Figure 1**. Part of the wild type ventral epidermis. (**A**) Ventral abdominal epidermis, including the central midline, showing rows of predenticles 1–6. (**B**) The denticles made later by the same larva. (**C**) A model: *ds* expression combined with expression of *fj*, gives the presumed pattern of Ds activity and explains the orientation of the rows 0–6. Each row points towards the neighbouring cell with the most Ds. The line within a cell (usually sloped) indicates that each cell has different amounts of Ds at its anterior and posterior limits (see *Figure 4*). In all the figures, one individual is imaged first at mid second stage and later after moulting to third stage. Note almost exact correspondence between predenticles and denticles in all cases. Cell boundaries (Ecad) are shown in red and actin highlighted in green (see 'Materials and methods'). Cells shaded in blue belong to the posterior compartment (*Lawrence and Struhl, 1996*). T1 and T2 indicate the two rows of tendon cells, shaded in grey in **C**. Although tendon cells do not show actin predenticles, they show characteristic actin palisade-like structures (*Saavedra et al., 2014*). The blue dotted lines are transects to locate the diagrammatic cross sections shown at the right of each figure. Anterior is to the left.

In *Figure 1C*, we show a hypothetical model of the segmental landscape of Ds activity in the epidermis of later larval stages. This model derives from mutant phenotypes of the third stage larvae (*Casal et al., 2006*; *Donoughe and DiNardo, 2011*) and experiments in which the distribution of Ds was manipulated, also at later stages (*Repiso et al., 2010*; *Donoughe and DiNardo, 2011*). The model also depends on the pattern of expression of *four-jointed* (Fj), a kinase that activates Ft and deactivates Ds (*Brittle et al., 2010*; *Simon et al., 2010*). *fj* is much more strongly expressed in the tendon cells than elsewhere—it should lower the activity of Ds in these cells—and graded in cells from rows 2 (high) to 4 (low) (Saavedra et al., in preparation). These pieces of evidence taken together argue for, but do not prove, the segmental landscape of Ds activity shown in *Figure 1C*. The hypothetical landscape can explain the orientation of all the denticle rows.

## Atypical cells and multipolarity

If the relevant cells of the larva (cells from row 0 to row 6 and including the two rows of tendon cells) were stacked in 10 parallel rows like the bricks in a wall (as in *Figure 1A*), our model would be a sufficient explanation for the polarity of all the cells. But in reality, the arrangement of the cells is less orderly. Consider the cells of row 4. A few of these cells are tilted from the mediolateral axis; they take up 'atypical' positions, contributing to two different rows of cells in the normal stack (one is shown in *Figure 2A,B*, shaded magenta and *Figure 2—figure supplement 1*). In such a cell, one portion occupies territory between a row 3 cell (in which Ds activity is medium) and a T2 cell (in which Ds activity is low). Thus, this portion of the atypical cell has neighbours exactly like an ideal row 4 cell and

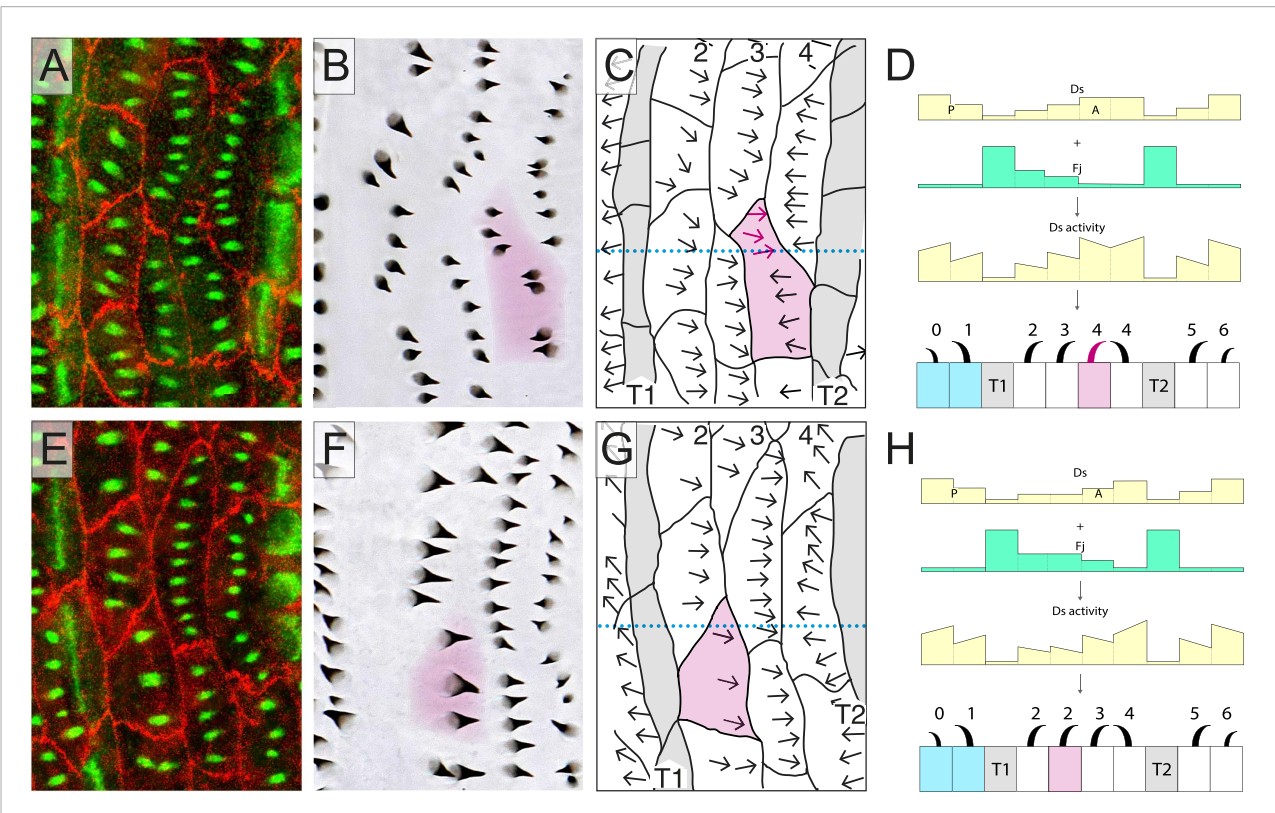

**Figure 2**. Atypical cells. (**A–D**) One atypical and multipolar cell, largely in row 4, is shown, in **B–D** (shaded in magenta). The transects shown as dotted lines in **C** and **G** are illustrated in **D** and **H** with the presumed amounts of Ds and Fj as well as the presumed activity of Ds. (**E–H**) One atypical cell of row 2 is shown; labelling as in other figures. See also *Figure 2—figure supplement 1*.

The following figure supplement is available for figure 2:

**Figure supplement 1**. Atypical cells: more examples.

its denticles point forwards towards the neighbouring row 3 cell (*Figure 2A–D* and *Figure 2—figure supplement 1*).

The neighbouring row 3 cell is presumed to have more Ds activity than the T2 cell (*Figure 2D* and *Figure 2—figure supplement 1*). However, the other portion of the same atypical cell intervenes between a row 3 and a normal row 4 cell and the denticles in that portion point backwards; again towards the neighbouring cell with higher Ds activity (in this case, a row 4 cell). Note that the backwards-pointing polarity adopted by this domain of the atypical cell does not, and is not expected to, affect the polarity of neighbouring cells. Its anterior neighbour, a row 3 cell, lies between a row 2 and a row 4, as does any normal row 3 cell, whereas its posterior neighbour, a row 4 cell, abuts a T2 cell that has a low Ds activity (a lower Ds activity than this portion of the atypical cell finds at its anterior interface). Therefore, under our hypothesis, cells touching this domain of the atypical row 4 cell do not differ, with respect to the Ds/Ft activities of their neighbours, from normal row 3 and 4 cells and consequently show normal polarity: thus, the row 3 cell points its denticles posteriorly, and the row 4 cell points its denticles anteriorly.

To quantitate, we selected atypical cells for study and then ask does the orientation of denticles in one part of a cell correlate with the anterior and posterior neighbours of that part? The answer is very clearly yes (*Table 1*). We explain below that these multipolar cells tell us that a portion of the membrane of one cell can compare itself with that in a facing portion of the same cell and this comparison polarises that particular domain of the cell. By this means a cell reads the Ds activities of its anterior and posterior neighbours and responds accordingly. In the case of the atypical row 4 cells, even though all their anterior neighbours are of the same type (row 3 cells), the neighbours at the posterior membrane are of two different types (T2 and row 4); accordingly, the two different regions of the cell manifest opposing polarities.

Atypical cells occur in other regions of the segment, for example in row 2. These cells have a mix of neighbours also, and anatomically are equivalent to the atypical cells near row 4; however, all their predenticles and denticles point backwards as they do in the wild type (*Figure 2E–H* and *Figure 2—figure supplement 1*). This fits exactly with the model (*Figure 2H* and *Figure 2—figure supplement 1*), because a cell, or part of a cell, that intervenes between T1

---

**Table 1**. Atypical cells: quantitation of denticle polarities in relation to neighbouring cells showing the effect of the Ds/Ft system

| Wild type | | | | $ds^-\ ft^-$ | | | |
|---|---|---|---|---|---|---|---|
| Anterior neighbour | Denticle polarity of atypical Row 2 cells† | | Posterior neighbour | Anterior neighbour | Denticle polarity of atypical Row 2 cells§ | | Posterior neighbour |
| | Anteriorly | Posteriorly | | | Anteriorly | Posteriorly | |
| T1 cell | 0 | **52** | Row 3 cell | T1 cell | 16 | 21 | Row 3 cell |
| Row 2 cell | 0 | **35** | Row 3 cell | Row 2 cell | 14 | 13 | Row 3 cell |
| Anterior neighbour | Denticle polarity of atypical Row 4 cells‡ | | Posterior neighbour | Anterior neighbour | Denticle polarity of atypical Row 4 cells¶ | | Posterior neighbour |
| | Anteriorly | Posteriorly | | | Anteriorly | Posteriorly | |
| Row 3 cell | **110** | 8* | T2 cell | Row 3 cell | 54* | 37 | T2 cell |
| Row 3 cell | 8** | **41** | Row 4 cell | Row 3 cell | 24 | 20** | Row 4 cell |

†Denticles of 29 atypical cells.
‡Denticles of 27 atypical cells. 8 of 8* (and 6 of 8**) were predenticles located in ambiguous positions and their denticles were arbitrarily allocated to those classes favouring the null hypothesis. Fisher exact test p-value: $<2.2^{-16}$.
§Denticles of 23 atypical cells. Fisher exact test p-value: 0.6135.
¶Denticles of 24 atypical cells. 1 of 54* (and 1 of 20**) were predenticles located in ambiguous positions and their denticles were arbitrarily allocated to those classes disfavouring the null hypothesis. Fisher exact test p-value: 0.7104.
Numbers in bold emphasise the main result of the table that is, the effect of neighbours on denticle polarities in the wildtype. This effect does not exist in the mutant larvae.

and row 3 (i.e., like a normal row 2 cell) makes backward-pointing denticles; they point away from the tendon cells (where Ds is low) and towards cells where the Ds activity is presumed to be higher. The situation is the same for a cell or part of a cell that inserts itself between another row 2 cell and a row 3 cell (because, according to the model of the segmental landscape of Ds activity, row 3 has a higher Ds activity than row 2 [*Figure 1C* and *Figure 2H*; Saavedra et al., in preparation]) and they therefore make backwardly oriented denticles. If we quantitate as before, again we find complete agreement between the orientation of the denticles and the presumed Ds level of the neighbours, although in this case all the denticles point backwards (*Table 1*).

Atypical cells also occur in row 3; these cells have portions inserted between a different row 3 and a row 4 cell and their denticles point backwards toward the neighbour cell with higher Ds activity, that is row 4 (not shown).

## Testing the role of the Ds/Ft system

Up to now, we have hypothesised that the orientations of all denticles both in the normal and the multipolar cells are primarily, or only, determined by the Ds/Ft system. To test this hypothesis, we first looked at larvae in which the Ds/Ft system is broken. That system depends on heterodimeric bridges made up of Ds and Ft molecules; these bridges cannot form without either or both of these proteins. Using $ds^-ft^-$ larvae, we found that atypical cells do occur both in rows 2 and 4 and are anatomically similar to those in wild type. However, the orientation of denticles in these cells is equally indeterminate in both rows and is independent of neighbours (*Figure 3A–D*, *Table 1*). This is presumably and simply because, without the Ds-Ft intercellular bridges, a cell cannot compare neighbours to ascertain where to point its denticles. It also argues that there is no other PCP agent, apart from the Ds/Ft system, that is responsible for these multipolar cells in the larva.

In a further test, we made small clones of marked cells that over-express *ds* and studied them in the larva in vivo. These clones were initiated early in embryogenesis, around the blastoderm stage and are small, ranging in size from 1 to 5 cells (*Saavedra et al., 2014*). They probably begin to make excess Ds in the embryo following the first one or two divisions that occur in the postblastoderm epidermis (*Foe and Alberts, 1983*; *Hartenstein and Campos-Ortega, 1985*) Consistent with observations on the adult of clones expressing high levels of *ds* (*Casal et al., 2002*), or on larvae when groups of cells over-express *ds* (*Repiso et al., 2010*; *Donoughe and DiNardo, 2011*), neighbouring wild type cells become repolarised so that their denticles point towards cells with excess Ds. In larvae, we now find that portions of cells are affected and the orientation of denticles in that part of the cell in contact with the *ds*-expressing cell can point inwards whilst other parts of the same cell, whose neighbours are not over-expressing *ds*, point in various directions. *Figure 3E–H* illustrates a nice example, several multipolar cells are induced by the clone and one entire cell is reoriented. In this and other examples, multipolar cells also appear occasionally to affect the polarity of their neighbouring cells (see legend, *Figure 3*). However, such a propagation of polarity seems to be limited to those portions of neighbouring cells that contact regions of the multipolar cells that show an abnormal polarity. Again, this finding supports our conclusion that polarity is the result of a local and independent comparison between facing membranes of cell domains. Note however that this local propagation of polarity does not affect row 4 cells. We believe that this observation can be explained if the low Ds activity of the T2 cells were to strongly influence the polarity of row 4 cells, strongly enough to counteract any effect that might originate from any multipolar or repolarised row 3 neighbours. Note that the marked *ds*-expressing cells themselves show no preferred polarity, their denticles pointing in diverse directions. This might be explained by the amount of Ds activity in such cells being much higher than all their neighbours, giving insufficient directional bias to fix their own polarities.

## Gaps in the tendon cells

A different atypical situation occurs occasionally when there is an opening in the usually continuous rows of tendon cells. Such gaps often affect the orientation and pattern of denticles, as well as forming some multipolar cells. Examples of these are shown (*Figure 3—figure supplement 1*), and they find the same general explanation. Consider first a cell that is located next to a gap in the T1 tendon row and thus adjacent to a row 1 and a row 3 cell. The rows 1 and 3 have similar amounts of Ds activity and so the cell in question develops no defined polarity (magenta in *Figure 3—figure supplement 1B–D*). In another case, we see the T2 row of tendon cells is interrupted and in the breach there is a denticulate cell that

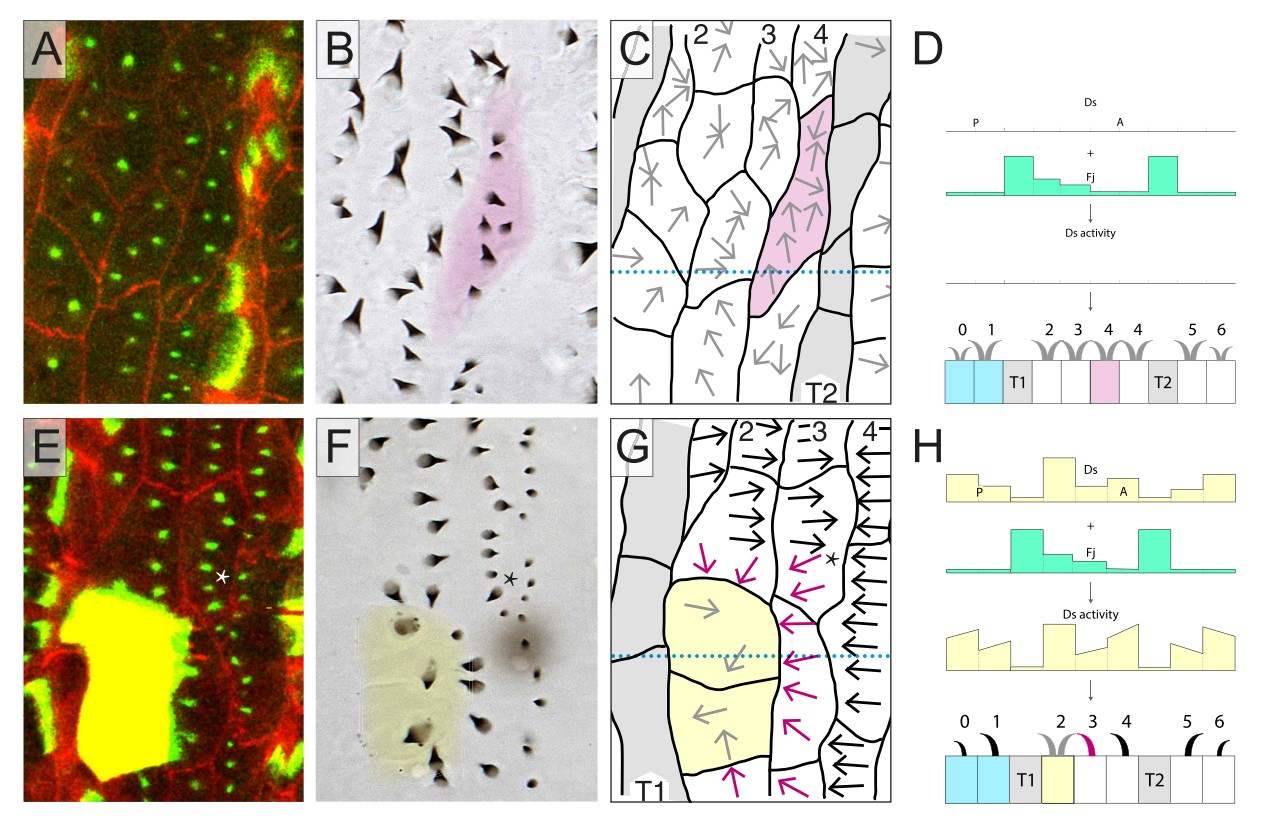

**Figure 3**. Testing the model. (**A–D**) An epidermis lacking both the *ds* and *ft* genes. The cell shaded in magenta is a row 4 cell which has an atypical disposition. The predenticle and denticle orientation is variable and awry. (**E–H**) A clone of two cells over-expressing *ds* (marked in yellow, **E** and shaded in yellow, **F–H**) imposes an orientation on all or parts of the neighbouring wild-type cells. Note that one reoriented cell (*) appears to have no direct contact with clonal cells expressing *ds* but only with neighbours of such cells. We have seen this in other cases. In the adult abdomen such propagation extends to several cell diameters (*Casal et al., 2002*; *Casal et al., 2006*) and has also been reported in larvae (*Repiso et al., 2010*; *Donoughe and DiNardo, 2011*). See also *Figure 3—figure supplement 1*.

The following figure supplement is available for figure 3:

**Figure supplement 1**. Breaches in rows of tendon cells.

comes to lie between a row 4 and a row 6 cell (*Figure 3—figure supplement 1E–H*). This creates an ambiguous situation as both these neighbours have similar amounts of Ds and give no directional cue. Consequently, the cell in question has predenticles and denticles that are placed in the middle of the cell (*Figure 3—figure supplement 1E*), giving rise to denticles that are spread over the apical surface of the cell and point in mixed directions (shown in magenta in *Figure 2—figure supplement 1F–H*). Again, both these situations fit with the model that cells or parts of cells point their denticles towards that neighbour cell that has the most Ds activity.

## Refining the model

These results offer evidence that the earlier models of the Ds/Ft system (*Strutt and Strutt, 2002*; *Casal et al., 2006*) apply to these large larval cells, just as they do to the eye (*Strutt and Strutt, 2002*; *Simon, 2004*), the wing (*Ma et al., 2003*), and the abdomen, (*Casal et al., 2006*). For example, in all these organs and stages, the cell membrane of a clone containing excessive amounts of Ds will attract Ft to the abutting membranes of neighbouring cells and lead to a redistribution of Ds-Ft heterodimers in those cells, and even beyond (reviewed in *Thomas and Strutt, 2012*).

We proposed (*Casal et al., 2006*) that the polarity of a cell is the outcome of a comparison between its anterior and posterior membranes; that the amount and orientations of Ds-Ft

heterodimers are compared and the denticles point towards that region of the cell membrane that has most Ft (i.e., abutting a neighbouring cell that has higher Ds levels). The multipolar cells provide new evidence for this hypothesis. First, note that the level of Ds in a row 3 cell should be higher than the level of Ds in a T2 cell but lower than the level of Ds in a row 4 cell (*Figure 1C*). Then, consider the following arguments: one that there is a comparison between the anterior and posterior membranes of the row 4 multipolar cell filled in magenta in *Figure 2B–C*. Two, that this comparison is local to different regions of that cell. Across the whole stretch of its anterior membrane this row 4 multipolar cell contacts two cells of row 3, each presenting equal amounts of Ds. However, within the posterior membrane of this same multipolar cell there are two separate regions, each abutting cells with different levels of Ds: a region abutting the T2 neighbour, and a region abutting the normal row 4 cell. The region abutting the T2 has a lower amount of Ft (as a result of the low amount of Ds in the T2 cell), and a higher amount of Ds (as a result of the high amount of Ft in the T2 cell) than the facing anterior membrane of the same cell, so its denticles point forwards. The region abutting the normal row 4 cell has a higher amount of Ft (as a result of the high amount of Ds in the normal row 4 cell) and a lower amount of Ds (as a result of the lower amount of Ft in the normal row 4 cell) than in the facing anterior membrane of the same cell, so its denticles point backwards.

Multipolarity tells us something new and unexpected: that the comparison is local to different regions of the cell. The two domains described above are comparing their facing anterior and posterior membranes, independent of each other. Therefore, the comparison cannot be a signal that pervades the whole cell, and instead multipolarity suggests the existence of oriented 'conduits' that link facing regions within the anterior and posterior membranes of a single cell. If such oriented conduits exist they could allow the directional transport and/or unequal stabilisation of components of the polarity machinery (such as Ds and Ft themselves). Our conception of these conduits is yet incomplete; but if their orientation were determined by the distribution of Ds and Ft, and if they also helped convey Ds and Ft across the cell, then together both these properties could constitute a feedback mechanism. Such a mechanism would make the polarity of cells more robust and also would affect the propagation of polarity from cell to cell. Understanding propagation is important because experiments suggest that the distribution of Ds and Ft within the membrane of one cell is partly determined by the distribution of these molecules in that cell's neighbours and also in cells beyond (reviewed in *Thomas and Strutt, 2012*).

The disposition of Ds-Ft heterodimers, as indicated in *Figure 4*, will be the result of these processes. In *Figure 4*, we imagine the distribution of Ft-Ds heterodimeric bridges in the epidermis and speculate in more detail how local comparisons might determine denticle polarity of the cells or parts of cells. Note that if Ds and Ft are transported across the cell between limited domains, there is no need to invoke free diffusion of these molecules (as in *Mani et al., 2013*; *Abley et al., 2013*). Similar conduits would be present in all cells, but normally, since they signal consistently in all parts of the cell, they would not be detected as separable elements.

There are hints at what these conduits might be: several authors described oriented microtubules in planar polarised cells (*Fristrom and Fristrom, 1975*; *Eaton et al., 1996*; *Turner and Adler, 1998*) and studied microtubule growth and polarity in the developing wing (*Harumoto et al., 2010*; *Olofsson et al., 2014*). In a particular part of the wing, and for a relatively short time, these authors noted that there are microtubules that are oriented proximodistally. In this part of the wing, they showed a slight preponderance (ca 5%) of oriented microtubules with the minus ends proximal and their plus ends distal and proposed that Ds and Ft are instrumental in this net orientation. Changing the distribution of Ds in a test part of the wing changed the net orientation of the microtubules (*Harumoto et al., 2010*). Our hypothesis of conduits could relate to these findings; microtubules could form all, part or none of these conduits. But, if they do it is not clear why, in most parts of the wing and for most of the time, the microtubules and the hairs are not co-oriented. In any case, a net orientation of microtubules might be read out as a net transport of vesicles from proximal to distal (*Shimada et al., 2006*; *Harumoto et al., 2010*; *Gault et al., 2012*). A similar process in the larval epidermal cells could lead to the subsequent orientation of predenticles in the membrane, but how they would do this is unknown. Microtubules have been observed in cells of the embryonic epidermis at the time that the denticles of the first larval stage

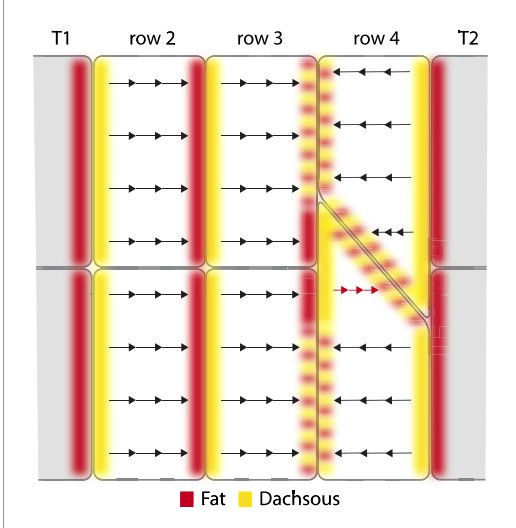

**Figure 4**. Polarised conduits in the cells. A hypothetical view of how Ds and Ft polarise cells or parts of cells. All membranes contain both kinds of dimers: these are Ds in the cell $x$ (Ds$^x$), linked to Ft in the neighbouring cell $y$ (Ft$^y$), or Ft in the cell $x$ (Ft$^x$) linked to Ds in the neighbouring cell $y$ (Ds$^y$). We proposed (*Casal et al., 2006*) that intercellular bridges consisting of heterodimers of Ds and Ft are asymmetrically distributed in a polarised cell and determine the polarity of that cell. In the diagram, we indicate a majority of Ds by yellow, and a majority of Ft by rufous. Some membranes contain similar numbers of Ds and Ft and these are shown with alternating blotches of the two colours. Arrows indicate the sign and the paths of the oriented conduits that span between facing and limited areas of membrane. Such conduits can give small parts of cells an individual polarity, as in the atypical cell in row 4 (red arrows). The tendon cells, T1 and T2, largely drive the segmental pattern of Ds activity—they have low Ds activity and therefore the majority of heterodimers formed between a T2 and a row 4 cell are Ds$^4$–Ft$^{T2}$. Similarly at the boundary between a row 2 and a row 3 cell, the majority of the heterodimers are Ft$^2$–Ds$^3$; partly because at the opposite boundary between a T1 and a row 2 cell, the heterodimers are largely Ft$^{T1}$–Ds$^2$. Where row 3 and row 4 cells meet in the wild type, they are imagined to have similar levels of Ds$^3$–Ft$^4$ and Ft$^3$–Ds$^4$ because, at that cell junction similar, but opposite, effects from very low Ds levels in both T1 and T2 tendon cells converge. However, in the red-arrowed region, at the anterior limit of the atypical cell, the heterodimers are mostly Ft$^3$–Ds$^4$. We propose that in this red-arrowed region, the deployment of heterodimers is the outcome of a different comparison made between facing subregions of the anterior and posterior limits of this atypical cell. What is different about this comparison? In this red-arrowed region, the cell's anterior neighbours (row 3) have less Ds than the posterior neighbour (a normal row 4). As a result, at the boundary between row 3 and atypical row 4, the heterodimers will be mostly Ft$^3$–Ds$^4$.

*Figure 4. continued on next page*

are formed, although their actual polarity is not known (*Price et al., 2006*; *Marcinkevicius and Zallen, 2013*).

## Multipolarity in other cell types

Motile cells such as fibroblasts or *Dictyostelium* tend to extend lamellipodia in different directions at once, net movement resulting if one direction is favoured over others (*Shi et al., 2013*). It is not clear if this kind of multipolarity relates to PCP: a defining feature of PCP is that the orientation of a cell is fixed by cell interaction and this is not usually the case with isolated motile cells. However, in plants, pavement cells are genuinely multipolar. They show that even between two identical neighbours, local interactions can be of different sign and can organise the cytoskeleton in different ways to build cells that, at least near the periphery, have regions of opposing polarities (*Xu et al., 2010*).

It has been thought that PCP in animals involves the whole cell and includes organelles and the cytoplasm, structures that form in membranes such as cilia or stereocilia as well as pervasive outputs such as mechanical tension (*Wallingford, 2010*; *Deans, 2013*; *Guillot and Lecuit, 2013*). Multipolarity in the larval cells of *Drosophila* that we report makes it clear that PCP is, or can be, subcellular; consequently, some current models of PCP may need to be adapted. For instance, our observations raise the possibility that all cells are fundamentally 'multipolar' but, usually, all subregions of a cell are subject to consistent polarising influences and are co-oriented.

## Materials and methods

### Mutations and transgenes

Flies were reared at 25°C on standard food. The Flybase (*St Pierre et al., 2014*) entries of the relevant constructs used in this work are the following: *DE-cad::tomato*: *shg$^{KI.T:Disc\RFP-tdTomato}$*; *sqh.utrp::GFP*: *Hsap\UTRN$^{sqh.T:Avic\GFP-EGFP}$*; *UAS. cherry::moesin*: *Moe$^{Scer\UAS.P\T.T:Disc\RFP-mCherry}$*; *UAS. stinger::GFP*: *Avic\GFP$^{Stinger.Scer\UAS.T:nls-tra}$*; *UAS. ectoDs*: *ds$^{ecto.Scer\UAS}$*; *UAS.cd8::GFP*: *Mmus \Cd8a$^{Scer\UAS.T:Avic\GFP}$*; *tub>stop>Gal4*: *P{GAL4-αTub84B(FRT.CD2).P}*; *sry.FLP*: *Scer\FLP1$^{sry-alpha}$*; *ds$^-$*: *ds$^{UA071}$*; *ft$^-$*: *ft$^{15}$*.

### Experimental genotypes

(*Figure 1*, *Figure 2*, *Figure 2—figure supplement 1*, *Figure 3—figure supplement 1*) *w; DE-cad:: tomato* sqh.utrp::GFP/ CyO-P{Dfd-EYFP}2.
(*Figure 3A–D*) *w; ds$^-$ ft$^-$ sqh.utrp::GFP/ ds$^-$ ft$^-$ DE-cad::tomato*.

*Figure 4. Continued*

However, at the facing boundary, between atypical and normal row 4, there are balancing influences from anterior and posterior directions, as in a boundary between a normal row 3 and a normal row 4, and as a consequence the amounts of $Ds^4$–$Ft^4$ and $Ft^4$–$Ds^4$ heterodimers should be similar.

(*Figure 3E–H*) *w; tub>stop>Gal4 UAS.cd8::GFP sqh.utrp::GFP/ UAS.stinger::GFP DE-cad::tomato; sry.FLP UAS.cherry::moesin/ UAS.ectoDs.*

## Handling and observation of larvae

Second stage larvae at the pre-third stage were mounted in a drop of Voltalef 10S oil on a microscope slide and imaged using a Leica SP5 confocal microscope. The larvae were carefully removed, kept at 25°C on an agar plate with fresh yeast paste until they moulted into the third stage; cuticles of third stage were prepared using a standard protocol (*Saavedra et al., 2014*).

For *Table 1*, predenticles of cells of rows 2 and 4 with atypical dispositions were classified as follows: predenticles localised in a domain of the cell that abutted a tendon cell (i.e., T1 or T2), predenticles localised in a domain that abutted a non-tendon cell (i.e., another row 2 or 4 cell), or predenticles localised in an intermediate domain. The orientations of the denticles formed by these predenticles were scored, one by one, in the third stage cuticles. Late second stage larvae with small clones of marked cells in the epidermis were obtained as previously described (*Saavedra et al., 2014*).

## Acknowledgements

We thank the Bloomington Stock Center, Marco Antunes, Marcus Bischoff, Paola Cognini, Thomas Lecuit, and Eurico Morais de Sá for flies and members of the lab for discussions. MR was partially supported by an Erasmus Placement *scholarship* and PS was supported by studentships from Fundação para a Ciência e a Tecnologia and the Cambridge Philosophical Society. This work was kindly supported by the Wellcome Trust Investigator Award to PAL, WT096645MA

## Additional information

### Funding

| Funder | Grant reference number | Author |
|---|---|---|
| Wellcome Trust | WT096645MA | Peter A Lawrence |

The funder had no role in study design, data collection and interpretation, or the decision to submit the work for publication.

### Author contributions

MR, PS, Conception and design, Acquisition of data, Analysis and interpretation of data; JC, PAL, Conception and design, Analysis and interpretation of data, Drafting or revising the article

### Author ORCIDs

José Casal, http://orcid.org/0000-0002-5149-1335

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
