## [Decision Letter]

Thank you for sending your work entitled “Planar cell polarity: regions within a single cell can be polarised independently” for consideration at *eLife*. Your article has been favourably evaluated by K VijayRaghavan (Senior editor and Reviewing editor) and 2 reviewers. As you can see the substantial comments are quite readily addressable.

The Reviewing editor and the reviewers discussed their comments before we reached this decision, and the Reviewing editor has assembled the following comments to help you prepare a revised submission.

Lawrence and coworkers have shown in previous work that the *Drosophila* larval epidermis serves as excellent model system to study the Ft/Ds pathway, in particular, as in larvae emerging denticles can be followed relative to individual cell borders over time. Depending on the row of cells in a segment, denticles either point anteriorly or posterior, respectively. In addition, in late stages (L3), there are two rows of tendon cells that lack denticles and serve as borders of Ds activity (as they apparently express high levels of Fj repressing Ds activity based on a paper under preparation and not available to the reviewer). Here, Rovira et al. propose a model that each cell compares Ds activity levels of its anterior and posterior neighbours and grows the denticle towards the neighbour with higher Ds activity. To test, they make use of the fact that the larval epithelium is not a perfect wall of bricks, but that some of the very large cells contact 2 cells of different rows on one of their sides (with anticipated distinct activities of Ds).

Rovira et al. show that such cells can point denticles in both orientations, of course not randomly as in Ft/Ds mutants, but according to the (hypothetical) Ds activity difference between the neighbours in contact at that position. This is highly intriguing. On the wings of *Drosophila* PCP mutants, cells frequently produce two hairs with divergent polarity, with the individual hairs matching the polarity of different neighbouring cells.

However, the embryonic multipolar cells described in the manuscript are wild-type and therefore more amenable to meaningful analysis. In retrospect, the striking results are perhaps not completely surprising as they fit older models in which Ds and Ft across a membrane can recruit each other to the boundary. Importantly, over-expression of Ds in a cell can repolarise cells (known), but do so for parts of a cell only that it is in contact with (leaving the rest of the cell polarised as instructed by neighbours; but see below), which is intriguing.

The Irvine (Ambegaonkar, 2012) and Strutt (Brittle, 2013) labs have made tagged Ds/Ft constructs that may be expressed in mosaics in the larva (the authors state that the epithelial cells divide after they can induce flip-pout clones; it thus may be possible to use their FLP to make mitotic clones). We do not ask that this be done for acceptance of the paper as a 'Research Advance' for *eLife*, but only suggest this as possible direction the authors may like to consider.

While the work presented finds astonishing differences of polarity within a single cell, a lot in the paper is based on the proposed model (without looking at Ft or Ds levels).

One concern is that the model shown in Figure 4 does not obviously match the mechanism described in the text (paragraph two of subsection headed “Refining the model” in the Results and Discussion section). While the model is just that and not merely a summary of the realists, it will serve all well if the authors re-craft the text to bring clarity and avoid concerns in about what the authors mean to convey.

Specifically, the Ds landscape shown in Figure 1 can produce the observed denticle pattern if each cell points its denticles towards the neighbouring cell with the highest Ds activity. A cell can sense the levels of Ds in neighbouring cells by monitoring the orientation of Ft-Ds dimers that are shared with each neighbour. In effect, a cell points its denticles towards the region of its own membrane that has the greatest proportion of Ft (rather than Ds) within Ft-Ds dimers.

However, in Figure 4, both the boundary between row 3 and 4 cells, and the atypical boundary between row 4 and 4 cells, are represented as having approximately equal amounts of Ft-Ds dimers in each orientation. If this were the case, how would the red arrow (oriented conduit) form in the atypical cell, since initially both 3/4 and 4/4 boundaries would have similar Ft-Ds orientations? For the model to work, the whole 3/4 boundary should have a majority (but not all) of Ft-Ds dimers with Ds in the row 4 cell.

Some further points about the model, which can be elaborated in the revised manuscript. It is not clear why the 3/4 boundary to the left of the red arrow has Ft-Ds dimers that are all oriented with Ds in the row 4 cell. Why does this differ from the rest of the 3/4 boundary? The figure legend states that “this deployment of heterodimers is the result of a local comparison made between facing subregions of the anterior and posterior limits of this atypical cell”. There is some (apparent?) circularity between cause and effect here. Our understanding is that the local comparison is between Ft and Ds distribution in the facing subregions, not that the comparison determines the distribution of Ft and Ds. Do clarify this in the revised manuscript.

Some more points:

Introduction section: “This comparison appears to involve limited regions of membranes that face each other along the anterior to posterior axis and depends on oriented conduits that span between the two membranes.” This sentence is confusing as it implies that ‘oriented conduits’ are required for the Ft/Ds comparison process, although my understanding is that the oriented conduits are proposed to result from the Ft/Ds comparison. In addition, the existence of ‘oriented conduits’ appears from this sentence to be established, whereas they are actually a proposal of the model. Please correct this.

---

## [Author Response]

*Lawrence and coworkers have shown in previous work that the Drosophila larval epidermis serves as excellent model system to study the Ft/Ds pathway, in particular, as in larvae emerging denticles can be followed relative to individual cell borders over time. Depending on the row of cells in a segment, denticles either point anteriorly or posterior, respectively. In addition, in late stages (L3), there are two rows of tendon cells that lack denticles and serve as borders of Ds activity (as they apparently express high levels of Fj repressing Ds activity based on a paper under preparation and not available to the reviewer). Here, Rovira et al. propose a model that each cell compares Ds activity levels of its anterior and posterior neighbours and grows the denticle towards the neighbour with higher Ds activity. To test, they make use of the fact that the larval epithelium is not a perfect wall of bricks, but that some of the very large cells contact 2 cells of different rows on one of their sides (with anticipated distinct activities of Ds)*.

*Rovira et al. show that such cells can point denticles in both orientations, of course not randomly as in Ft/Ds mutants, but according to the (hypothetical) Ds activity difference between the neighbours in contact at that position. This is highly intriguing. On the wings of Drosophila PCP mutants, cells frequently produce two hairs with divergent polarity, with the individual hairs matching the polarity of different neighbouring cells*.

We are interested in knowing where this point is published, as if it is true we ought to and would like to refer to it, can the reviewer oblige with a reference?

*However, the embryonic multipolar cells described in the manuscript are wild-type and therefore more amenable to meaningful analysis. In retrospect, the striking results are perhaps not completely surprising as they fit older models in which Ds and Ft across a membrane can recruit each other to the boundary. Importantly, over-expression of Ds in a cell can repolarise cells (known), but do so for parts of a cell only that it is in contact with (leaving the rest of the cell polarised as instructed by neighbours; but see below), which is intriguing*.

*The Irvine (Ambegaonkar, 2012) and Strutt (Brittle, 2013) labs have made tagged Ds/Ft constructs that may be expressed in mosaics in the larva (the authors state that the epithelial cells divide after they can induce flip-pout clones; it thus may be possible to use their FLP to make mitotic clones). We do not ask that this be done for acceptance of the paper as a 'Research Advance' for* eLife*, but only suggest this as possible direction the authors may like to consider*.

Yes, thank you. We are aware of these constructs and had already begun to explore in this direction and have been making new genetic stocks. The discovery in this paper of multipolar cells can, we think, be exploited to learn more but obviously that will take time.

*While the work presented finds astonishing differences of polarity within a single cell, a lot in the paper is based on the proposed model (without looking at Ft or Ds levels)*.

Yes, we are sorry about this. The model is evidenced in another experimental paper (Saavedra et al., in preparation) that is not yet complete. We have been doing some difficult experiments to complete it and the hope is that it should be finished and submitted to a specialised journal within a couple of months. We didn't want to wait to publish our discovery of multipolar cells, nor to “bury” this discovery in a long and detailed paper about denticle patterns.

*One concern is that the model shown in*
Figure 4
*does not obviously match the mechanism described in the text (paragraph two of subsection headed “Refining the model” in the Results and Discussion section). While the model is just that and not merely a summary of the realists, it will serve all well if the authors re-craft the text to bring clarity and avoid concerns in about what the authors mean to convey*.

Okay, see below.

*Specifically, the Ds landscape shown in*
Figure 1
*can produce the observed denticle pattern if each cell points its denticles towards the neighbouring cell with the highest Ds activity. A cell can sense the levels of Ds in neighbouring cells by monitoring the orientation of Ft-Ds dimers that are shared with each neighbour. In effect, a cell points its denticles towards the region of its own membrane that has the greatest proportion of Ft (rather than Ds) within Ft-Ds dimers*.

Agreed.

*However, in*
Figure 4*, both the boundary between row 3 and 4 cells, and the atypical boundary between row 4 and 4 cells, are represented as having approximately equal amounts of Ft-Ds dimers in each orientation. If this were the case, how would the red arrow (oriented conduit) form in the atypical cell, since initially both 3/4 and 4/4 boundaries would have similar Ft-Ds orientations? For the model to work, the whole 3/4 boundary should have a majority (but not all) of Ft-Ds dimers with Ds in the row 4 cell*.

This description reveals that we have not explained the model in Figure 4 adequately. Therefore, we have expanded and hopefully improved the arguments that lead to Figure 4 both in the text and in Figure 4 legend.

*Some further points about the model, which can be elaborated in the revised manuscript. It is not clear why the 3/4 boundary to the left of the red arrow has Ft-Ds dimers that are all oriented with Ds in the row 4 cell. Why does this differ from the rest of the 3/4 boundary? The figure legend states that “this deployment of heterodimers is the result of a local comparison made between facing subregions of the anterior and posterior limits of this atypical cell”. There is some (apparent?) circularity between cause and effect here. Our understanding is that the local comparison is between Ft and Ds distribution in the facing subregions, not that the comparison determines the distribution of Ft and Ds. Do clarify this in the revised manuscript*.

Thank you, we have tried to do this.

*Some more points*:

*Introduction section: “This comparison appears to involve limited regions of membranes that face each other along the anterior to posterior axis and depends on oriented conduits that span between the two membranes.” This sentence is confusing as it implies that ‘oriented conduits’ are required for the Ft/Ds comparison process, although my understanding is that the oriented conduits are proposed to result from the Ft/Ds comparison. In addition, the existence of 'oriented conduits' appears from this sentence to be established, whereas they are actually a proposal of the model. Please correct this*.

We do hypothesise that the “oriented conduits” are polarised in consequence of a comparison of the facing membranes (across the cell) and also are the agents that compare facing membranes. And we have tried to be clearer about what we mean by causes and effects.